# Biodegradable Nanohybrid Materials as Candidates for Self-Sanitizing Filters Aimed at Protection from SARS-CoV-2 in Public Areas

**DOI:** 10.3390/molecules27041333

**Published:** 2022-02-16

**Authors:** Anton M. Manakhov, Elizaveta S. Permyakova, Natalya A. Sitnikova, Alphiya R. Tsygankova, Alexander Y. Alekseev, Maria V. Solomatina, Victor S. Baidyshev, Zakhar I. Popov, Lucie Blahová, Marek Eliáš, Lenka Zajíčková, Andrey M. Kovalskii, Alexander N. Sheveyko, Philipp V. Kiryukhantsev-Korneev, Dmitry V. Shtansky, David Nečas, Anastasiya O. Solovieva

**Affiliations:** 1Research Institute of Clinical and Experimental Lymphology—Branch of the ICG SB RAS, 2 Timakova st., Novosibirsk 630060, Russia; permyakova.elizaveta@gmail.com (E.S.P.); sitnikovanat9@gmail.com (N.A.S.); 2Research Laboratory of Inorganic Nanomaterials, National University of Science and Technology “MISiS”, Leninsky Prospekt 4, Moscow 119049, Russia; andreykovalskii@gmail.com (A.M.K.); sheveyko@mail.ru (A.N.S.); kiruhancev-korneev@yandex.ru (P.V.K.-K.); shtansky@shs.misis.ru (D.V.S.); 3Nikolaev Institute of Inorganic Chemistry SB RAS, 3 Acad. Lavrentiev Ave., Novosibirsk 630090, Russia; alphiya@yandex.ru; 4Research Institute of Virology, The Federal Research Center of Fundamental and Translational Medicine, 2 Timakova st., Novosibirsk 630060, Russia; al-alexok@ngs.ru (A.Y.A.); mariaza@ngs.ru (M.V.S.); 5Department of Computer Engineering and Automated Systems Software, Katanov Khakas State University, Pr. Lenin 90, Abakan 655017, Russia; bayd_vs@mail.ru; 6Emanuel Institute of Biochemical Physics RAS, Kosygina 4, Moscow 119334, Russia; zipcool@bk.ru; 7Central European Institute of Technology CEITEC-BUT, Purkyňova 123, 61200 Brno, Czech Republic; lucie.blahova@ceitec.vutbr.cz (L.B.); marek.elias@ceitec.vutbr.cz (M.E.); lenkaz@physics.muni.cz (L.Z.); yeti@physics.muni.cz (D.N.)

**Keywords:** SARS-CoV-2, nanofibers, antiviral coating, plasma, XPS, silver, copper

## Abstract

The COVID-19 pandemic has raised the problem of efficient, low-cost materials enabling the effective protection of people from viruses transmitted through the air or via surfaces. Nanofibers can be a great candidate for efficient air filtration due to their structure, although they cannot protect from viruses. In this work, we prepared a wide range of nanofibrous biodegradable samples containing Ag (up to 0.6 at.%) and Cu (up to 20.4 at.%) exhibiting various wettability. By adjusting the magnetron current (0.3 A) and implanter voltage (5 kV), the deposition of TiO_2_ and Ag^+^ implantation into PCL/PEO nanofibers was optimized in order to achieve implantation of Ag^+^ without damaging the nanofibrous structure of the PCL/PEO. The optimal conditions to implant silver were achieved for the PCL-Ti0.3-Ag-5kV sample. The coating of PCL nanofibers by a Cu layer was successfully realized by magnetron sputtering. The antiviral activity evaluated by widely used methodology involving the cultivation of VeroE6 cells was the highest for PCL-Cu and PCL-COOH, where the VeroE6 viability was 73.1 and 68.1%, respectively, which is significantly higher compared to SARS-CoV-2 samples without self-sanitizing (42.8%). Interestingly, the samples with implanted silver and TiO_2_ exhibited no antiviral effect. This difference between Cu and Ag containing nanofibers might be related to the different concentrations of ions released from the samples: 80 μg/L/day for Cu^2+^ versus 15 µg/L/day for Ag^+^. The high antiviral activity of PCL-Cu opens up an exciting opportunity to prepare low-cost self-sanitizing surfaces for anti-SARS-CoV-2 protection and can be essential for air filtration application and facemasks. The rough cost estimation for the production of a biodegradable nanohybrid PCL-Cu facemask revealed ~$0.28/piece, and the business case for the production of these facemasks would be highly positive, with an Internal Rate of Return of 34%.

## 1. Introduction

The recent COVID-19 pandemic has shown that pathogens can spread rapidly across the world, having a catastrophic impact on the health of human beings. As no single solution preventing the spreading of viral infections exists, multiple-barrier protection mechanisms are required to block or at least slow down the speed of virus transmission [1]. Although numerous efforts to slow down the spreading of the virus have been developed, including vaccines [2], diagnostic technologies and developments in surveillance measures for SARS-CoV-2-positive patient contact tracing [3], the rapid growth of new patients has not slowed down, and additional measures are necessary to combat the pandemic crisis.

Since the transmission of viral infections often occurs through aerosol (often considered the most dangerous, as it can spread infectious particles in high titers) [4], the filtration and separation of submicron-sized contaminants are one of the main objectives of modern nanotechnology. Ultra-thin fibers obtained by the electrospinning process have shown great potential in the application of these materials as active filter layers due to their unique physical and chemical properties, namely low basic weight, small pore size, high permeability, high specific surface (from 1 to 100 m^2^/g, depending on the fiber diameter and intra-fiber porosity), good interconnection of pores and potential for incorporating active chemical species or functionalization at the nanoscale [5]. These materials can be used to protect against aerosol nanoparticles; chemicals (such as nerve agents and mustard gas) and biological threats, including bacterial spores, viruses, etc. The ability of ultrafine fiber filters to effectively filter particles larger than 10 nm has previously been demonstrated, making them suitable for a wide range of filtration applications, including the use of nanofibers in masks and respirators.

The filtration efficiency increases when the fiber diameter and distance between fibers are decreased to the nanometer scale due to the increased contact probability between aerosol particles and fiber surface. Moreover, the slip flow effect diminishes the friction between aerosol particles and the ultra-thin fibers, leading to a small pressure drop. 

Omori et al. made high-efficiency air filters with a hybrid nanofiber/microfiber structure using wet paper processing. The hybrid filters exhibited high performance with a goodness Qf = 0.043 (Qf is a widely used index indicating filtration efficiency due to pressure drop) for ultrafine fibers with an average diameter of 180 nm while filtering 100-nm particles [6].

Skaria et al. compared the filtering performance of commercially available masks with a nanofiber-based filter prototype (Secure Fit, PT) under development [7]. They showed that the nanofiber filter was able to filter more effectively than commercial masks. They showed that, unlike commercial facemasks, the prototype nanofiber filter produced significantly reduced the mask’s resistance to airflow and resulted in more exhaled air from the facemask.

Another problem with filter media is secondary contamination, where stopped viral and bacterial particles migrate in the filter layer, reaching the respiratory organs. This can be solved either by sterilizing the filter for the reuse or creation of self-cleaning materials.

Most nanofiber filters, such as polyacrylonitrile (PAN) [8], poly(ε-caprolactone) (PCL) [9] and poly(vinylidene fluoride) (PVDF) [10], withstand water washing and sterilization with alcohol. Recently, Ullah et al. [11] compared the cleaning efficiency of a surgical mask filter with a nanofiber filter in 75% ethanol to evaluate their reusability. Unlike surgical masks, the filtering mechanism of the nanofiber filter does not depend on static charge but is based on structural characteristics such as pore size and distribution. Since the nanofiber morphology was not affected by disinfection, the nanofiber-based filters also retained their ability to effectively filter as they did prior to use, unlike surgical masks.

Lee et al. developed a fiber filter formed from a polybenzimidazole (PBI) solution with high a filtration efficiency (PM2.5~98.5%) at a significantly reduced pressure drop of 130 Pa [12]. This filter showed thermal stability after hot plate treatment at 400 °C for 1 h and retained its original performance after several cleaning cycles. The O_2_ nanomask produced by Viaex Technologies can be sterilized by a washing machine [13].

However, the development of self-sanitizing surfaces by introducing antibacterial and antiviral coatings or materials is highly desired [14,15]. Antibacterial coatings and materials are widely studied and employed, while the antiviral or virucidal properties of the materials are less known [16]. While bacteria are single-celled living organisms, viruses are not considered to be ‘alive’ due to their reliance on a host to reproduce and survive [1]. Nevertheless, many materials have both antiviral and antimicrobial properties, even though the efficacy can differ.

The persistence of viruses on surfaces depends on materials properties (porosity, hydrophobicity, roughness, etc.); physical factors (temperature, humidity, etc.) and biological factors (structure of the virus or presence of microbial biofilm on a surface) [17,18]. Therefore, developing a material with precisely selected properties for the targeted application is crucial while creating antiviral surfaces. For example, if one is looking to develop a self-sanitizing facemask to filter the viruses for an infinite period, it is essential to absorb as many viruses as possible and destroy their structure by virucidal agents. In contrast, self-sanitizing layers for everyday surfaces would instead require antiadhesive properties combined with similar virucidal agents. Thus, as superhydrophobic surfaces stimulate the absorption of viruses (due to interactions between the hydrophobic outer surface of proteins and solid surfaces), the filters for facemasks should preferably possess a hydrophobic nature, while self-sanitizing surfaces should be superhydrophilic [19]. As for porosity, viruses, including SARS-CoV-2, can survive for a more extended period on a porous surface. An active virus was found on a facemask after seven days, while no virus was found on smooth surfaces after the same period.

Virucidal (antiviral) agents can be based on different substances: metal ions or oxides, peptides, organic zwitterions, etc. [1,5,20]. Various metals, including silver, gold, copper, zinc and others, have antibacterial and antiviral properties. The biocidal properties of silver are well-documented, and the popularity of Ag-based antibacterial agents lies in its low toxicity and biocidal effect at low concentrations [21,22]. Although the antibacterial effect mechanism of both copper and silver is well-known and thoroughly investigated, their virucidal properties are less-explored. It is assumed that the virucidal effect of these metals can be based on the destruction of the viral RNA genome (genomic damage) or membrane disruption. It was shown that the virucidal properties of copper rely primarily on the release of copper ions. Both Cu(II) and Cu(I) ions contribute to virus inactivation.

Additionally, the reactive oxygen species produced by silver or copper may also destroy viruses. The antipathogenic contact killing/inactivating performance of copper cold spray surfaces and coatings was reviewed in Reference [23]. Mantlo et al. showed that the Luminore CopperTouch surfaces inactivated 99% of SARS-CoV-2 in 2 h [20]. Tremiliosi et al. [24] studied the activity of polycotton tissues modified with silver nanoparticles against Gram-positive bacteria, Gram-negative bacteria, fungi and SARS-CoV-2 virus. It was shown that the Ag-modified tissue showed 99.99% activity against fungi and both types of bacteria and 99.6% activity against the SARS-CoV-2 virus.

Minoshima et al. evaluated the antiviral activity of different copper and silver compounds and demonstrated that the antiviral mechanism of these metals against influenza viruses is mediated by the inactivation of hemagglutinin (HA) and neuraminidase (NA) surface proteins of the viruses [25]. The action of Cu_2_O significantly differed from that of other ionic copper and silver compounds and showed a substantial inactivation of viruses with and without an envelope. The authors concluded that utilizing inorganic chemicals containing copper and silver as anti-influenza materials would potentially reduce the risk of viral transmission in the environment. A high potential of these materials in exploring a combination of copper and silver with biocidal coating chemicals such as photocatalytic TiO_2_ nanoparticles indicates that using Cu_2_O to treat both public and living spaces may help limit or even prevent the spread of viruses. Indeed, the combination of different virucidal and bactericidal agents (so-called nanohybrid composites) have a great potential to boost the virus deactivation [26], because state-of-the-art self-sanitizing coatings require at least two hours to inactivate SARS-CoV-2, which can be considered as too slow of an inactivation process.

Moreover, the presented problem of standard facemask protection efficiency can be solved by applying novel nanomaterials, including biodegradable nanofibers with multiple virucidal agents. Indeed, at present, a large volume of polypropylene facemasks (that should be considered a biohazard) are disposed of every day, while they are not capable of stopping the spreading of the virus. At the same time, the scientific community has already started looking for materials for facemasks with a high degree of protection, as summarized in several works [3,18,27,28,29,30].

This work developed novel biodegradable nanohybrid materials consisting of PCL or PCL/PEO nanofibers coated by a Cu layer or TiO_2_ layer decorated with Ag nanoparticles, as schematically shown in Figure 1. Furthermore, we tested the Cu-coated PCL covered by COOH plasma polymer layer containing reactive carboxylic groups. We explored biodegradable nanohybrid materials’ antiviral properties against SARS-CoV-2. Our approach was based on a scalable, environment-friendly and economically viable robust technology consisting of the electrospinning of nanofibers and their coating by magnetron sputtering, and thus, it has a high potential for future commercialization.

## 2. Results

### 2.1. Simulation of Ag^+^ Implantation into PCL Matrix

The implantation of silver ions into a very soft matrix such as the polymer has to be carefully adjusted. Thus, first of all, we studied the penetration of Ag^+^ ions into the PCL matrix (as it is the main component of PCL/PEO mixed nanofibers and for the sake of simplicity with assumed nearly similar results for pure PCL and PCL/PEO 75:25 hybrid nanofibers).

The PCL unit cell was taken from Reference [31] and relaxed in VASP; after which, it was used to create a slab supercell.

To simulate the irradiation of a film of finite thickness, a PCL slab with a size of 22.7 nm × 3.94 nm × 5.31 nm, consisting of 64,800 atoms, was constructed. Periodic boundary conditions were applied in the direction of the y and z axes, and the irradiation was carried out along the x-axis. Before irradiation, the slab was relaxed for 100 ps at constant pressure (NPT thermostat), then at a constant temperature (NVT thermostat) equal to T = 300 K.

The metal atom was placed randomly at a distance of 1.8 nm from the PCL slab surface, mainly in the center of the YZ plane of the supercell (Figure 2). The atom was given an initial velocity component normal to the slab plane in accordance with the energy under consideration. In the simulation, a variable time step was used, which was selected from the condition that the maximum displacement of atoms did not exceed 0.001 nm. This procedure made it possible to avoid unreasonably large movements of atoms and kept the simulation stable. For example, at the most significant energies of the metal atom considered, the minimum step was 0.012 fs.

To simulate the dissipation of energy into an infinite volume of material, a 0.5-nm-thick region was isolated from the side of the PCL slab to the atoms of which temperature control was applied (NVT thermostat, T = 300 K). An NVE thermostat was applied to the remaining atoms of the system, including the metal atom. The simulation continued while the energy of the metal atom exceeded the average thermal energy of the PCL slab.

The energies of the incident Ag atom from 500 eV to 2400 eV were considered. A series of computer experiments was made for each selected energy, consisting of five simulations. The average values are shown in Figure 3.

In the energy range under consideration, a power law dependence of the penetration depth on the atom’s energy is obtained, described by the linear equation L = 9.0 × 10^−3^ E in contrast to the power equation (L = 4.07 × 10^−2^ E^0.7739^) in Reference [32] for Cu penetration to PCL. During the penetration into the PCL, the metal atom can change the movement direction due to collisions with polymer atoms, the penetration angle of the silver atom is random and the average deviation angle for all simulations is 2.86 degrees, which are two times smaller in comparison with Cu atom [32] due to atomic weight difference.

### 2.2. Morphology of Biodegradable Nanohybrid Materials

The SEM micrographs of PCL/PEO-Ti-Ag nanofibers are presented in Figure 4. It can be seen that Ag^+^ implantation at 15 kV (Figure 4b) led to a significant destruction of nanofibrous material as compared to PCL/PEO-ref. The samples prepared at 5 and 8 kV exhibited morphology similar to the pristine PCL/PEO-ref. The Cu-coated PCL nanofibers (PCL-Cu) exhibited no changes (not shown here). The variation of fibers diameters was evaluated from the SEM image analysis and shown in Figure 5. It is evident that the mean values for all nanofibers (except the PCL/PEO-Ti0.5-Ag15kV) were similar to the reference values of the pristine nanofibers. The PCL/PEO-Ti0.5-Ag15kV sample was not analyzed because of the significant degradation of fiber morphology (Figure 4b).

### 2.3. Chemical Characterization of Biodegradable Nanohybrid Materials

The PCL and PCL/PEO compositions are well-known and reported elsewhere [33,34]. The modified nanofibers are of much higher interest, and the compositions of all the samples evaluated by XPS are summarized in Table 1. The incorporation of TiO_2_ and Ag in the PCL/PEO-Ti-Ag samples and significant incorporation of Cu in PCL-Cu were already evident from the surface’s overall atomic composition. However, understanding the chemical nature of titanium and silver requires a detailed analysis of Ti2p and Ag3d XPS spectra.

XPS Ti2p signal of the Ti layer is shown in Figure 6a. Due to the better-resolved Ti2p 3/2 signal as compared to the Ti2p 1/2 counterpart, the first peak was used for further XPS analysis. The XPS Ti2p 3/2 peak was fitted using three components (Figure 6a): TiC (BE = 455.3 eV, FWHM = 1.2 eV), TiN (BE = 456.6 eV FWHM = 1.8 eV) and TiO_2_ (BE = 458.5 eV, FWHM = 1.2 eV), indicating the presence of carbide, nitride and oxide states on the Ti surface. As shown before, the structure of the Ti2p spectrum for a layer deposited in such conditions does not vary with the current values [35,36].

The implantation of Ag ions after the titania deposition led to significant changes of the Ti2p environment. After Ag^+^ implantation at 15 kV, the nanofiber surface was fully covered by the titanium oxide layer composed of TiO_2_ (major component, BE = 458.4 eV, FWHM = 1.1 eV) and TiO (minor component, BE = 457.3 eV, FWHM = 1.9 eV), as shown in Figure 6b. A similar phase composition of the Ag ion-modified layer was seen in PCL-Ti0.3-Ag8kV and PCL-Ti0.3-Ag5kV (Figure 6c,d). Hence, the implantation of Ag ions leads to significant changes in the titania layer. A similar effect of titanium oxidation upon Ag^+^ implantation was found before for layers deposited at significantly higher ion energies and less sensitive substrates [37].

The implantation at 15 kV also led to a significant degradation of the C1s spectrum (Figure 7b), which can be seen from the comparison of C1s signal coming from PCL-Ti0.5-Ag15kV with a spectrum of PCL/PEO-ref. The C1s spectrum of PCL/PEO-ref was fitted with a sum of three components: CH_x_ (BE = 285 eV, FWHM = 1.1 eV), C–O (BE = 286.4 eV, FWHM-1.1 eV) and C(O)O (BE = 289 eV, FWHM = 1.1 eV). The spectrum of PCL/PEO-Ti0.5Ag-15kV exhibited very high FWHM for CH_x_ equal to 2.0 eV. The C(O)O contribution disappeared, and a new component attributed to C=O (BE = 288.3 eV, FWHM = 1.7 eV) was observed. Hence, despite the very high concentration of Ti (18 at.%) and relatively significant concentration of Ag (0.6 at.%), this sample should not be selected, as its nanostructure during the deposition process was not preserved, as revealed by SEM.

The deposition of the Ti layer at lower current and Ag^+^ implantation at 8 and 5 kV allowed to preserve the nanofibrous structure and avoid significant degradation of the carbon environment. As shown in Figure 7c,d, the PCL-Ti0.3-Ag8kV and PCL-Ti0.3-Ag5kV spectra were similar to PCL/PEO, with only small differences in the carbon environments. The main difference between these two samples was the intensities of the Ag signals. As shown in Figure 8a and Table 1, the concentration of Ag was highest in the PCL-Ti0.3-Ag5kV sample. All silver atoms were bonded to oxygen in Ag_2_O, as it can be concluded from the symmetrical shape of the Ag 3d 5/2 and Ag 3d 3/2 lines and the position of BE for Ag 3d 5/2 of 368.2 eV [38].

The Cu layers were deposited onto PCL nanofibers, and this process was thoroughly investigated elsewhere [32]. The C1s signal of the as-prepared PCL nanofibers (PCL-ref) can be fitted by the sum of four components, namely hydrocarbons CH_x_ (BE = 285 eV), carbon neighbored to ester group C-C(O)O (BE = 285.5 eV), ether group C-O (BE = 286.4 eV) and ester group C(O)O (BE = 289.0 eV). The FWHM for all peaks was 1.0 ± 0.1 eV. The XPS C1 signal of PCL-Cu (Figure 7f) was fitted using three components: CH_x_, C-O and C(O)O, where CH_x_ was the dominating environment. The COOH plasma polymer layer almost entirely covered the PCL-Cu-COOH samples, as only 0.4 at.% of Cu was detected by XPS (Table 1). The composition of PCL-Cu-COOH was similar to the surface chemistry reported previously [32,37,39].

The copper environment in the PCL-Cu sample (Figure 8b) was analyzed by fitting Cu2p 3/2 solely (without the Cu2p 1/2) due to the higher signal of the Cu2p 3/2 line. The XPS Cu2p 3/2 signal was fitted by a sum of six components: metallic copper or copper oxide (I) Cu^0^/Cu^+^ (BE = 932.5 ± 0.1 eV, FWHM = 1.1 ± 0.1 eV); copper oxide (II) CuO (BE = 934.5 ± 0.1 eV, FWHM = 1.7 ± 0.2 eV); copper hydroxide Cu(OH)_2_ (BE = 935.8 ± 0.2 eV FWHM = 1.9 ± 0.2 eV) and three Cu(II) satellites centered at 940.3, 941.4 and 943.8 eV.

### 2.4. Ag^+^ Ion Release

The kinetics of the silver ion release of the studied samples differed significantly (Figure 9). In sample PCL-Ti0.3-Ag5kV, there was a rapid release of silver during the first 24 h (12 μg/L), after which the release rate of silver ions decreased to 1 to 2 μg/L/day. The experimental data were fitted with the following function: $\left(1-exp\left(\sqrt{t/T}\right)\right)$, with *T* = 67 h, although it is unclear why this particular function gave the best fit for PCL/PEO-Ti0.3-Ag5kV. In samples PCL-Ti0.5-Ag15kV and PCL-Ti0.3-Ag8kV, the time dependence of the silver ion concentration in the solution had an approximately linear character, which is associated with a gradual release of silver ions from the depth of the sample. This means that even if the release of the silver ions follows an exponential function for these samples, the characteristic times *T* are significantly longer than 160 h.

### 2.5. Wettability

The wettability of all samples was evaluated by measuring the water contact angle (WCA). Optical images of droplets on the sample surfaces are shown in Figure 10. The PCL-ref sample exhibited very high WCA (~117°), and the deposition of the Cu layer led to the increased hydrophobicity of the sample (~133°). By depositing the COOH layer, the WCA decreased to ~10° due to the very hydrophilic nature of the COOH plasma polymer layer.

The WCA of PCL/PEO (with 75% PCL and 25% PEO) was ~108°. The deposition of titanium dioxide coating led to a significant decrease of WCA, and the implantation of Ag ions slightly improved the hydrophilic nature of the layer, but the deviations were not significant (Figure 10).

Hence, we succeeded in preparing nanofibrous samples with different ions implanted and various wettability that is highly interesting for investigating the antiviral effect.

### 2.6. Antiviral Tests

In vitro antiviral activity of the PCL-ref, PCL-Cu, PCL-Cu-COOH and PCL/PEO-Ti0.3-Ag-5kv samples against SARS-CoV-2 was estimated by the TCID50 assay. After incubation VeroE6 cells with SARS-CoV-2 and materials or without materials (control) for 3 days supernatant with viruses titrated on VeroE6 cells, which were cultivated on 96-well plates and determined the typical cytopathic effect (CPE) by using optical microscopy images (Figure 11) and the MTT test (Figure 12). The LgCPE50 was also determined for the all samples (Table 2). The most significant cytotoxic effect on the VeroE6 cell culture was observed in the control SARS-CoV-2 sample and for the PCL-ref and PCL/PEO-Ti0.3-Ag-5kv samples. In these two cases, a complete destruction of the monolayer of VeroE6 cells with the formation of apoptotic bodies and cell detritus on the 6th day was observed (Figure 11). Samples PCL-Cu and PCL-Cu-COOH had a cytoprotective effect, which was expressed in the preservation of the monolayer, adhesive activity of cells with the appearance of apoptotic changes in cells. These data indicate the antiviral activity of PCL-Cu and PCL-Cu-COOH materials (Figure 11). The MTT test results in a dilution of 10° show that, in the presence of the PCL-ref, PCL-Cu and PCL-Cu-COOH materials, the cell viability was higher compared with the VeroE6 cell culture infected with the SARS-CoV-2 virus strain without samples. This indicates a more pronounced metabolic activity and cell viability compared to the infected cells (Figure 12). When comparing the MTT test indicators for prototypes in the presence of the SARS-CoV-2 virus strain in dilutions of 10^−1^ and 10^−2^, no significant differences were found (data not shown).

Thus, our PCL/PEO nanofibers with implanted Ag^+^ ions exhibited no antiviral effect in contrast to the PCL-Cu and PCL-Cu-COOH layers. This difference can probably be attributed to the different loading of the silver ions compared to copper.

## 3. Discussion

In order to compare our results with the state-of-the-art data, we prepared a brief overview of antiviral Ag-based films and presented them in Table 3. Alshabanah and et al. [40] reported that 2–4% of Ag nanoparticles (NPs) containing PVA or TPU nanofibers have an antiviral effect; moreover, increasing the concentration of Ag NPs leads to a more significant decrease of remaining activity of blocking viral fusion of SARS-CoV-2.

Ju et al. demonstrated that nanofibers containing 4.29 wt% Ag NPs did not cause any significant reduction in virus titer after 15 min but a significantly decreased virus titer after 30 min and continued to reduce it even after 60 min.

Szymanska et al. [41] reported that incubation with AgNP-based hydrogels reduced HSV-1 attachment to cell surfaces by about 80–90% after 24 h. A substantial reduction in plaque numbers (above 60%) of HSV-2 was also observed.

Seino et al. [42] found that the antiviral activity strongly depended on the EMEM (Eagle’s Minimal Essential Medium) concentration. In textile fabrics modified with Ag NPs, influenza viruses were not detected after 2 h of contact at an EMEM concentration below 1/10. However, antiviral activity was suppressed under high EMEM concentrations. The results with Feline calicivirus showed the same tendency. Moreover, comparing the antiviral activity against the influenza A virus and Feline calicivirus, they concluded that the metallic Ag NPs were less effective to the nonenveloped viruses than enveloped counterparts.

It was demonstrated that, after 24 h of exposure, no infectious FCV was recovered when in contact with AgNP films, while the MNV titers decreased by 0.86 log TCID50/mL [43]. As for other natural compounds, Ag films exhibited the strongest antiviral effect at 37 °C [44].

According to Reference [32], Cu^2+^ released from PCL-Cu is about 80 µg/L/day (for incubation in the 50-mL volume), while in our study, the Ag ion release was ~15 µg/L/day. More intensive Ag^+^ ion implantation was not possible due to the degradation of the nanofibrous structure at a high voltage. In contrast, Cu coating by magnetron sputtering enabled a high dosage of Cu ions without damaging the nanofibrous structure, and it can be further increased (if necessary). Moreover, the high antiviral efficacy of PCL-Cu-COOH can be further enhanced by the grafting of active compounds blocking the activity of the viruses.

**Table 3 molecules-27-01333-t003:** Antiviral efficiency of Ag-contained materials.

Type of Material	Polymer	Concentration of Ag	Virus	Ref
nanofibers	Polyvinyl alcohol (PVA) Thermoplastic polyurethane (TPU)	2, 4wt%	SARS-CoV-2	[40]
nanofibers	Polyamide6	4.29 wt%	PDCoV	[45]
hydrogel	Carbopol 974P	25 ppm/100g 50 ppm/100g	Herpes simplex virus (HSV)-1,2	[41]
Textile	cotton	1,46 wt%	Influenza A Feline calicivirus	[42]
microfibers	poly (3-hydroxybutyrate-co-3-hydroxyvalerate) (PHBV)	2 wt%	feline calicivirus (FCV) murine norovirus 27 (MNV)	[43]

Implementing a biodegradable nanohybrid will depend on the economic aspects of proposed filters and facemasks. In order to roughly estimate the economic viability, we evaluated the cost structure for the preparation of biodegradable nanohybrid facemask using the available experimental data (consumption of chemicals, copper target and utilities). Open-source information for the pricing of solvents, PCL, copper targets, electrospinning equipment and magnetron sputtering machines were acquired from Alibaba.com (accessed on 29 December 2021). The CAPEX for purchasing a small electrospinning roll-t o-roll and magnetron sputtering machine manufactured in China is ~$380,000. Based on this evaluation, the cost of one piece of surgical mask made from biodegradable nanohybrid material was $0.28/piece, including the normalized capital cost (for the project lifetime of 10 years). The cost breakdown is shown in Figure 13. The price of advanced nanofibrous facemasks (but without antiviral properties) on eBay.com (accessed on 29 December 2021) varies between $1 and $2 per piece.

In order to evaluate the economic efficiency, we estimated the net present value and internal rate of return (IRR) for possible cases with the price of facemask $1 and $2 and the weighted average cost of the capital (WACC) 5 and 10%. The value-added tax of 20% was included in the calculations. In the most favorable case (WACC = 5% and price of the facemask = $2); the NPV and IRR were $4,122,000 and 84%, respectively. In the least favorable scenario (WACC = 10%, price of the facemask = $1), the NPV and IRR were $790,000 and 34%, respectively. These economic parameters can be considered highly attractive, especially in the era of negative rates in the European Central Bank. Indeed, this estimation must be further adjusted for the cases of actual production lines and must be “landed” for a selected country.

## 4. Materials and Methods

### 4.1. Electrospinning of PCL and PCL/PEO Nanofibers

The overall scheme for preparing antibacterial and antiviral nanofibers is depicted in Figure 1. The electrospun PCL nanofibers were prepared by electrospinning a 9 wt.% solution of polycaprolactone PCL (80,000 g/mol). The processing of the sample can be found elsewhere [34]. Briefly, the granulated PCL was dissolved in a mixture of acetic acid (99%) and formic acid (98%). All compounds were purchased from Sigma Aldrich (Darmstadt, Germany). The weight ratio of acetic acid (AA) to formic acid (FA) was 2:1. The PCL solutions in AA and FA were stirred at 25 °C for 24 h.

According to our methodology reported elsewhere [33], PCL/PEO nanofibers with a ratio 3:1 PEO prepared by electrospinning of the 9 wt.% solutions containing 75% PCL and 25% PEO (Mw = 100,000, Sigma-Aldrich Steinheim am Albuch, Germany).

The solutions were electrospun with a 20-cm-long wired electrode using a Nanospider™ NSLAB 500 machine (ELMARCO, Liberec, Czech Republic). The applied voltage was 50 kV. The distance between the electrodes was set to 100 mm. The as-prepared and nontreated PCL nanofibers are referred to as PCL-ref throughout the text.

### 4.2. Magnetron Sputtering of Cu

The Cu coatings were deposited by magnetron sputtering of a copper target in an ultra-high-vacuum deposition chamber (BESTEC, Germany). The input power of the magnetron was set to 37 W. Before the deposition, the chamber was evacuated to 6.2 × 10^−8^ mbar. A 30-sccm flow of a high purity Ar gas (99.99%) was introduced into the deposition chamber, setting the operation pressure to 1.5 × 10^−3^ mbar. During the film deposition, the target to substrate distance was 30 mm, and the substrate holder was rotated at 10 rpm to obtain a homogenous film thickness. The deposition time was adjusted to deposit a 50-nm-thick film (controlled by deposition onto Si wafer). The Cu-coated nanofibers are referred to as PCL-Cu throughout the text.

### 4.3. Deposition of TiO_2_ Coating and Ag Ion Implantation

The deposition experiments were performed using a vacuum set-up with a magnetron sputtering unit equipped with the MEVVA-type ion implanter [35,46]. The titania films (~20 nm thick) were deposited by magnetron sputtering of a composite TiC-CaO-Ti_3_PO_x_ target in a gaseous mixture of Ar and 15% N_2_. The applied magnetron current was 0.5 or 0.3 A, the magnetron voltage was ~450 V and the bias voltage was kept at −50 V (samples denoted as PCL/PEO-Ti). The target-to-substrate distance was fixed to 120 mm. Silver ions were implanted using a MEVVA-type implanter operating with the acceleration voltages of 15, 8 and 5 kV and the current of 20 mA. Hereafter, these samples are denoted as PCL/PEO-Ti0.5-Ag15kV for Ti current of 0.5A and Ag ion implantation voltage of 15 kV; PCL/PEO-Ti0.3-Ag-8kV and PCL/PEO-Ti0.3-Ag5kV for Ti current of 0.3 A and voltages of 8 kV and 5 kV, respectively [46].

### 4.4. Plasma COOH Coating

The COOH plasma polymer layers were deposited using a vacuum system UVN-2M equipped with rotary and oil diffusion pumps. The plasma was ignited using radio frequency (RF) power supply Cito 1310-ACNA-N37A-FF (Comet, Flamatt, Switzerland) connected to an RFPG-128 disk generator (Beams & Plasmas, Moscow, Russia installed in the vacuum chamber. The duty cycle and the RF power were set to 5% and 500 W, respectively. The residual pressure of the reactor was below 10^−3^ Pa.

CO_2_ (99.995%), Ar (99.998%) and C_2_H_4_ (99.95%) gases were fed into the vacuum chamber. The gas flows were controlled using a Multi-Gas Controller 647C (MKST, Newport, RI, USA). The flow rates of Ar, CO_2_ and C_2_H_4_ were set to 50, 16.2 and 6.2 sccm, respectively. The pressure in the chamber was measured by a VMB-14 unit (Tokamak Company, Dubna, Russia) and D395-90-000 BOC Edwards controllers. The distance between the RF electrode and the substrate was set to 8 cm. The deposition time was 15 min, and it led to the growth of ~100-nm-thick plasma coatings. The plasma-coated PCL-Cu nanofibers are referred to as PCL-Cu-COOH throughout the text.

### 4.5. Chemistry and Morphology Analysis

The microstructures of nanofibers and deposited layers was studied by scanning electron microscopy (SEM) using a JSM-7600F Schottky field emission scanning electron microscope (JEOL Ltd., Tokyo, Japan) equipped with an energy-dispersive X-ray (EDX) X-Max 80 Premium detector (Oxford Instruments, Abingdon, UK) operated at 15 kV.

The chemical composition of sample surfaces was determined by X-ray photoelectron spectroscopy (XPS) using an Axis Supra spectrometer (Kratos Analytical, Manchester, UK) equipped with a monochromatic Al Kα X-ray source. The maximum lateral resolution of analyzed area was 0.7 mm. The spectra were fitted using CasaXPS software after subtracting the Shirley-type background. The binding energies (BE) for all carbon and oxygen environments were taken from the literature [47,48]. The BE scale was calibrated by setting the CH_x_ component at 285 eV.

### 4.6. Modeling of Ag Atom Irradiation of PCL Surface

The classical molecular dynamics method in the LAMMPS [49] software package was applied to the simulations of PCL irradiation by Ag atoms. All interatomic interactions in the system were described by ReaxFF potentials [50]. The dimer energies were calculated by the selected potential (see Table 4) to estimate the parameters of the interaction of silver atoms with polymer atoms. The comparisons were made with similar calculations by the DFT method [50,51] in the Vienna Ab initio Simulation Package (VASP) [52,53,54] using the PBE exchange correlation functional. Despite that the ReaxFF potentials underestimate the energies of individual dimers, they qualitatively describe changes in the energy of interactions of Ag atoms with Ag, C, H and O, since the energy decreases from Ag-Ag to Ag-O both in the case of DFT and in the case of ReaxFF calculations. In addition, the difference in bond lengths between DFT and ReaxFF is negligible.

### 4.7. Ion Release and Wettability Measurements

Samples (PCL/PEO-Ti0.5-Ag15kV, PCL/PEO-Ti0.3-Ag-8kV and PCL/PEO-Ti0.3-Ag-5kV) with a size 10 × 10 mm^2^ were immersed in 50 mL of deionized water at room temperature for 1, 3 and 6 and 24 h, 3, 5 and 7 days to measure the release of Ag^+^ ions from the modified nanofibers. The concentrations of Ag ions in the collected deionized water were determined by inductively coupled plasma mass spectrometry (ICP-MS) using a X- Series II spectrometer (Thermo Fisher Scientific, Waltham, MA, USA).

The sample wettability was assessed by measuring the water contact angle (WCA). The measurements were carried out on an Easy Drop Kruss (Kruss, Germany) device. For each sample, at least five WCA measurements were performed.

### 4.8. Antiviral Tests

The in vitro study of antiviral activity of samples PCL-ref, PCL-Cu, PCL-Cu-COOH and PCL/PEO-Ti0.3-Ag-5kv (with maximum silver content determined by XPS analysis), was carried out in the culture of VeroE6 cells (АТСС ССL81, was taken from the National Collection of Smorodintsev Research Institute of Influenza, Russian Federation). This cell line was chosen because it is a highly expressing ACE2 receptor, which is the host target for SARS-CoV-2 and is commonly used for the determination of antiviral agents [55,56].

VeroE6 cells were seeded into a 24-well culture plate (TPP, Trasadingen, Switzerland) and cultured in DMEM/F12—Dulbecco’s Modified Eagle’s Medium with modification F12 (Capricorn, Germany) containing 10% fetal bovine serum (Capricorn, Ebsdorfergrund, Germany) and 100-IU/mL gentamicin at 37 °C in a 5% CO_2_ atmosphere for 24 h. After cultivation, the cell monolayer was washed twice with Hank’s solution (Biolot, Saint-Petersburgh, Russia). PCL-ref, PCL-Cu and PCL-Cu-COOH samples were sterilely placed to a monolayer of VeroE6 cells at a distance of 1 to 2 mm from the cell monolayer. A sustaining medium DMEM/F12 (Capricorn, Germany) with 2% fetal bovine serum (Capricorn, Germany) and 100 IU/mL gentamicin containing the SARS-CoV-2 virus strain at a dilution of 100 was placed in the wells and incubated at 37 °C for 72 h in a 5% CO_2_ atmosphere. Untreated noninfected cells and untreated virus-infected cells were used as negative and positive controls of infection, respectively. A VeroE6 cell culture infected with the SARS-CoV-2 strain in a dilution similar to the experimental samples was used as a control of the virus action. A visual assessment of the viability of the cell monolayer was carried out under an inverted microscope (Zeiss PrimoVert, Oberkochen, Germany,). Cell viability was assayed by the MTT method using the MTT Cell Proliferation Kit (Roche Diagnostics, Mannheim, Germany). The solution optical density was measured with a plate reader ELx808 (BioTek Instruments Inc., Winooski, VT, USA) at the wavelength of 540 nm. For each test sample and controls, independent measurements of the solution optical density were carried out during the MTT test. For statistical analysis, mean values and their standard deviations were calculated.

### 4.9. Techno-Economic Assessment

The cost estimation for the preparation of the PCL-ref samples was based on the consumption of the electrospinning solution evaluated in the lab, i.e., 20 mL per one sample (20 cm × 20 cm). The solvent and polymer costs were employed from the data available in alibaba.com. The acetic and formic acids’ price was 0.7 and 0.5 $/kg, while the price of PCL was 6.5 $/kg. The resulting price of electrospinning solution was $0.024 per one facemask. The copper target per 1 facemask was estimated from the lifetime of Cu target: one target is enough to deposit 12,000 nm coating onto the 20 cm × 20 cm sample. The price of 1 target was $50. The consumption of electricity was 0.25 kWh per 1 piece. The price of electricity was considered as 0.1 $/kWh.

In order to estimate CAPEX were used open-source data for electrospinning and magnetron sputtering equipment. The NPV and IRR were calculated using MS Excel embedded functions. The cost of electrospinning machine Nano Fiberlabs (Foschan, China) was taken from the available options at alibab.com and equaled $80,000. The cost of the magnetron sputtering machine was considered as $300,000 (HC VAC, Dongguan City, Guangdong Province, China).

## 5. Conclusions

We prepared a wide range of nanofibrous biodegradable samples containing Ag and Cu and exhibiting various wettability. The deposition of TiO_2_ followed by Ag^+^ implantation into PCL/PEO nanofibers was thoroughly investigated and optimized in order to achieve implantation of Ag^+^ without damaging of the nanofibrous structure of the PCL/PEO materials. The optimal conditions for silver ion implantation were achieved for PCL/PEO coated by TiO_2_ at the current of 0.3 A followed with the Ag^+^ ion implantation at a voltage of 5 kV, as it allowed to introduce 0.6 at.% of silver without damaging the nanofibrous structure of the sensitive PCL/PEO mixed nanofibers. The coating of PCL nanofibers by Cu layer was successfully performed by magnetron sputtering. The antiviral activity was the highest for PCL-Cu and PCL-COOH samples, and no antiviral activity was found for samples coated by TiO_2_ with implanted silver. This difference between Cu and Ag containing nanofibers might be related to the different ions released from the samples. The high antiviral activity of PCL-Cu opens an exciting opportunity to prepare low-cost, self-sanitizing surfaces for anti-SARS-CoV-2 protection and can be highly essential for air filtration application and facemasks. The rough estimation of the cost structure for PCL-Cu production revealed that manufacturing of one facemask would cost $0.28. If these PCL-Cu facemasks were sold at $1/piece, the Net Present Value and Internal Rate of Return would reach $790,000 and 34%, respectively. Hence, the production of biodegradable nanohybrid facemasks also has high commercial potential.

## Figures and Tables

**Figure 1 molecules-27-01333-f001:**
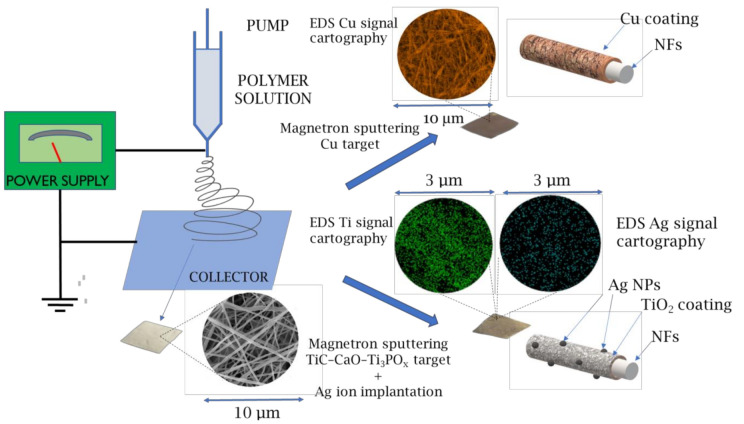
Scheme of the multistep process for antibacterial and antiviral nanofibers (NFs) preparation based on the electrospinning of polymers, magnetron sputtering of Cu or TiC-CaO-Ti_3_PO_x_ targets and implantation of silver. SEM-EDX mapping (in colored circles) is shown for the qualitative representation of synthesized materials.

**Figure 2 molecules-27-01333-f002:**
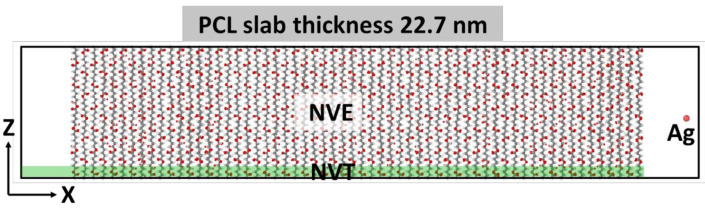
A diagram of the Ag^±^ implantation model with characteristic dimensions.

**Figure 3 molecules-27-01333-f003:**
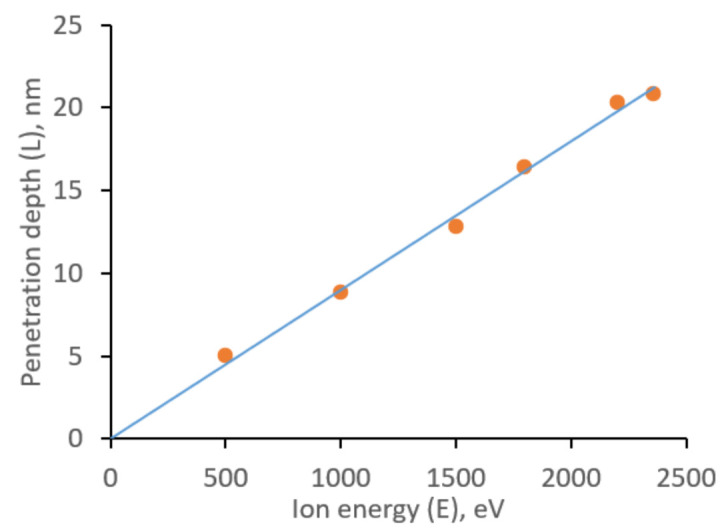
Dependence of the penetration depth of the Ag atom into PCL on its initial energy.

**Figure 4 molecules-27-01333-f004:**
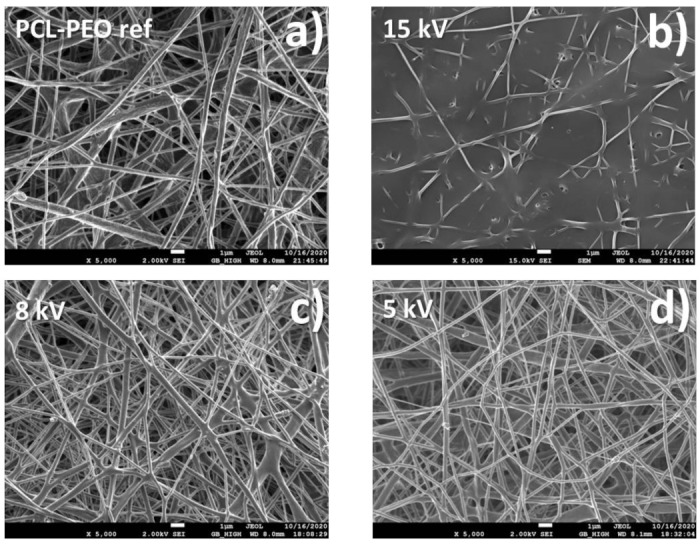
SEM micrographs of samples PCL/PEO-ref (**a**), PCL/PEO-Ti0.5-Ag15kV (**b**), PCL/PEO-Ti0.3-Ag-8kV (**c**) and PCL/PEO-Ti0.3-Ag-5kV (**d**).

**Figure 5 molecules-27-01333-f005:**
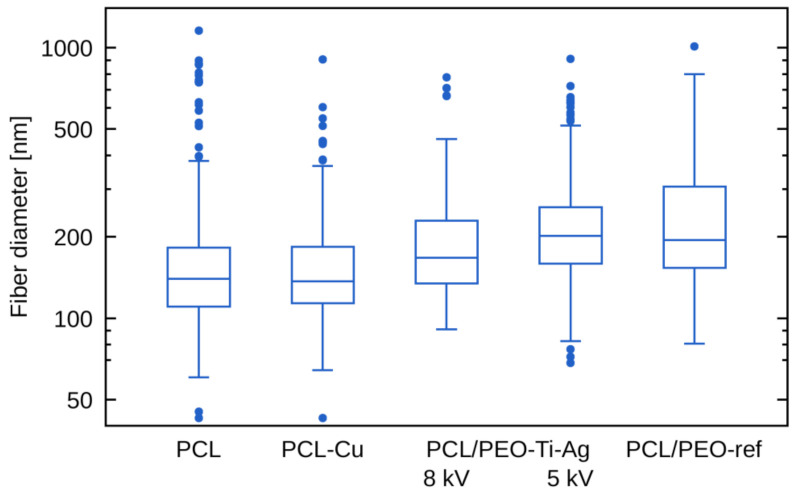
The fiber diameters for different samples evaluated from the SEM image analysis.

**Figure 6 molecules-27-01333-f006:**
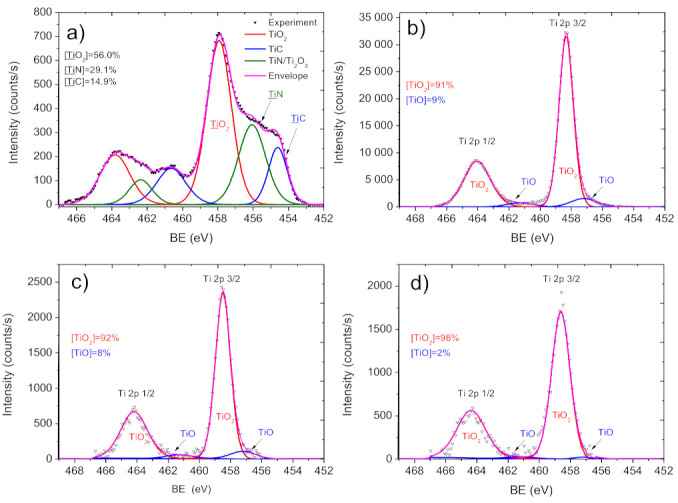
XPS Ti2p spectra and their curve fitting for samples PCL/PEO-Ti0.5 (**a**), PCL/PEO-Ti0.5-Ag15kV (**b**), PCL/PEO-Ti0.3-Ag5kV (**c**) and PCL/PEO-Ti0.3-Ag5kV (**d**).

**Figure 7 molecules-27-01333-f007:**
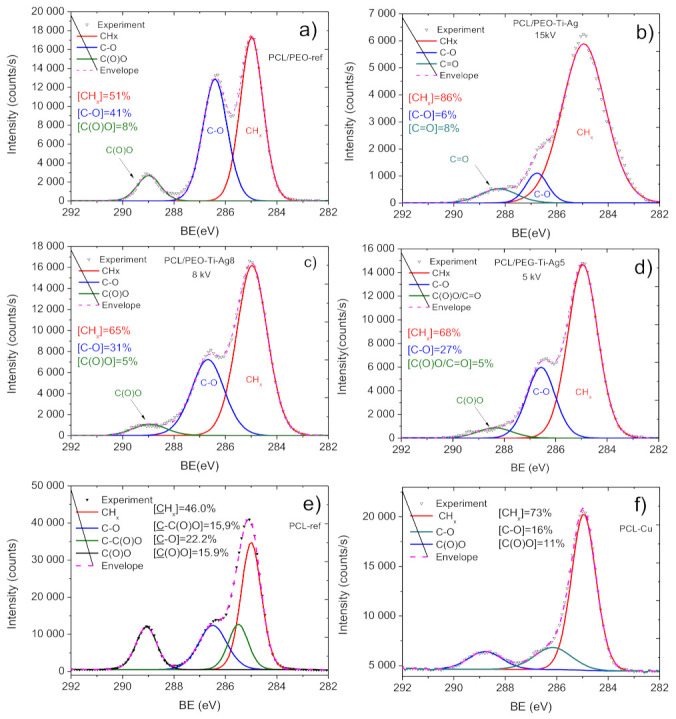
XPS C1s spectra and their curve fitting for samples PCL/PEO-ref (**a**), PCL/PEO-Ti0.5-Ag15kV (**b**), PCL/PEO-Ti0.3-Ag5kV (**c**), PCL/PEO-Ti0.3-Ag5kV (**d**), PCL-ref (**e**) and PCL-Cu (**f**).

**Figure 8 molecules-27-01333-f008:**
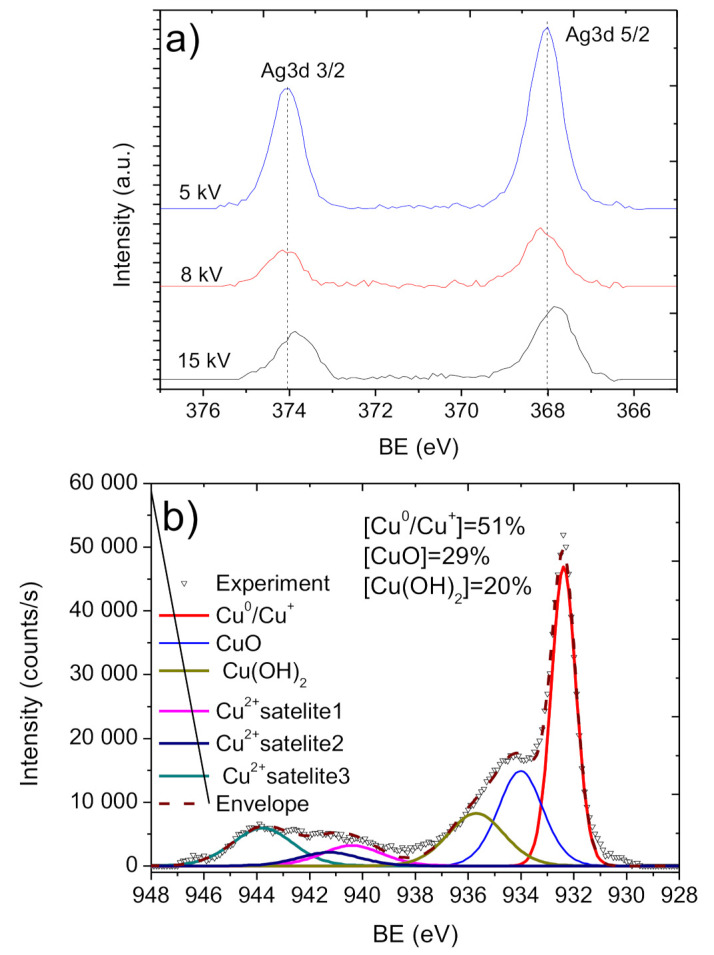
XPS Ag3d (**a**) spectra of PCL-Ti0.5-Ag15kV, PCL-Ti0.3-Ag8kV and PCL-Ti0.3-Ag5kV samples and Cu2p (**b**) spectrum of the PCL-Cu material.

**Figure 9 molecules-27-01333-f009:**
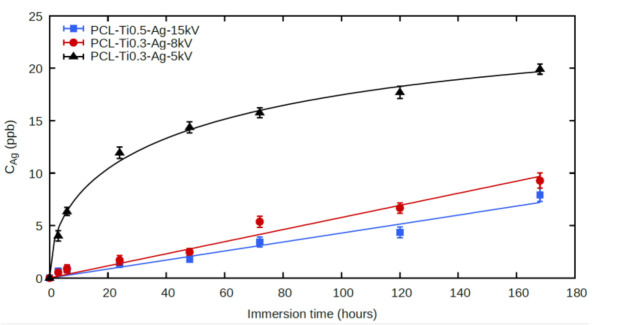
ICP MS results for Ag^+^ release from PCL/PEO-Ti0.3-Ag5kV, PCL/PEO-Ti0.3-Ag8kV and PCL/PEO-Ti0.3-Ag15kV samples.

**Figure 10 molecules-27-01333-f010:**
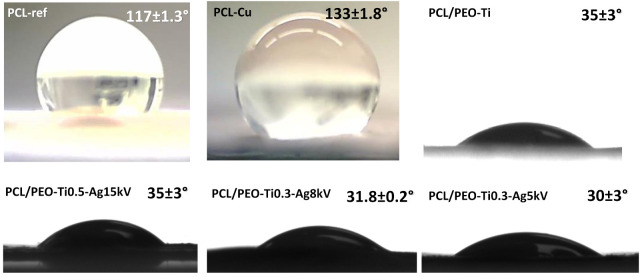
Optical images of droplets on the surface of the PCL-ref, PCL-Cu, PCL/PEO-Ti, PCL/PEO-Ti0.5-Ag15kV, PCL/PEO-Ti0.3-Ag8kV and PCL/PEO-Ti0.3-Ag5kV samples with corresponding WCA values.

**Figure 11 molecules-27-01333-f011:**
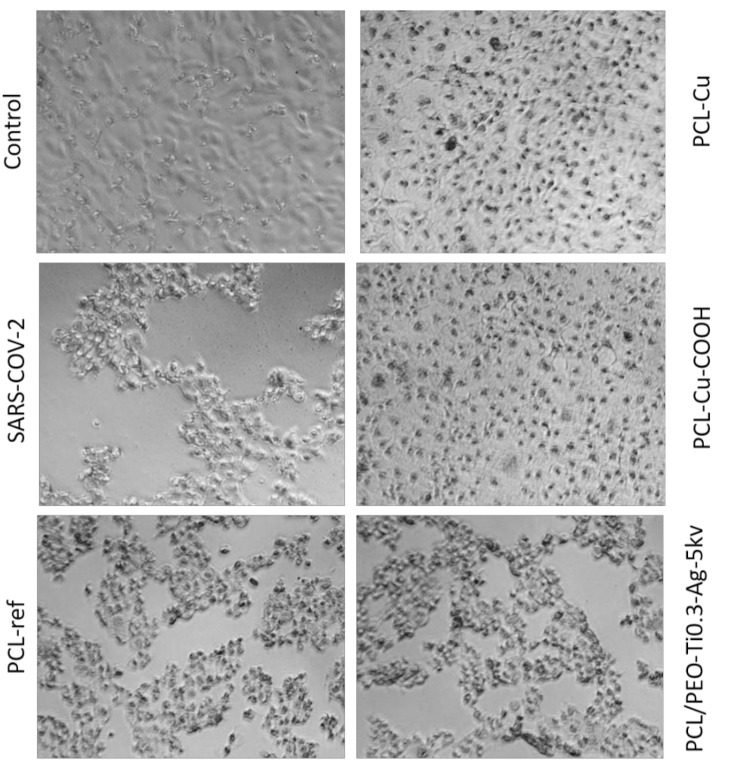
Optical micrographs of VeroE6 cells exposed to a virus for three days. In the presence of PCL-Cu and PCL-Cu-COOH materials, a significant protective (antiviral) effect is visible.

**Figure 12 molecules-27-01333-f012:**
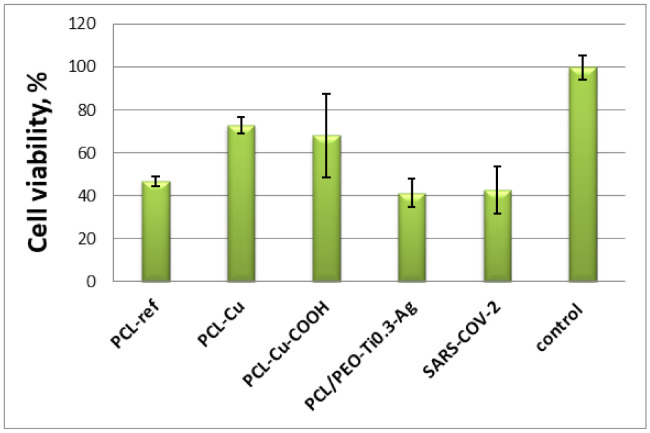
MTT assay of VeroE6 cells infected with SARS-CoV-2 (after incubation with samples), showing an antiviral activity of PCL-ref, PCL-Cu, PCL-Cu-COOH and PCL/PEO-Ti0.3-Ag-5kv materials and controls of the intact SARS-CoV-2.

**Figure 13 molecules-27-01333-f013:**
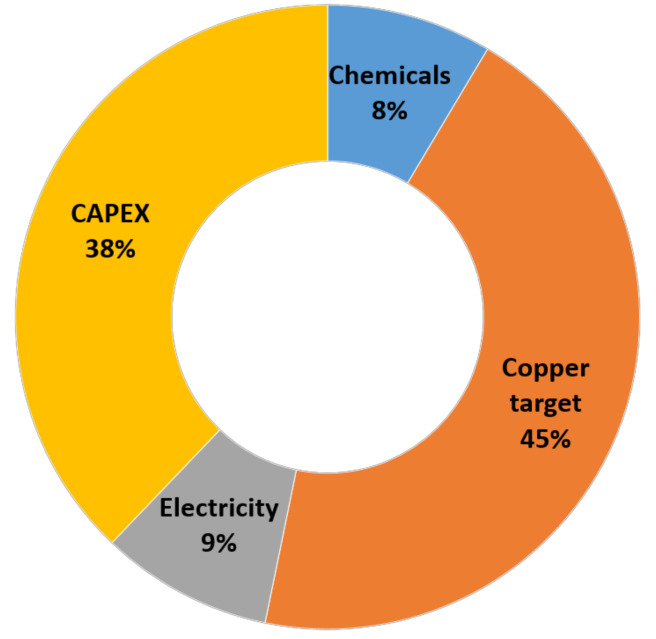
Cost structure of PCL-Cu facemask manufacturing.

**Table 1 molecules-27-01333-t001:** Composition of the samples (in at.%) derived from XPS analysis.

Sample	C (at.%)	O (at.%)	Ti (at.%)	Ag (at.%)	Cu (at.%)
PCL/PEO-ref	75.0	25.0	0.0	0.0	0.0
PCL/PEO-Ti0.5-Ag15kV	31.9	49.5	18.0	0.6	0.0
PCL/PEO-Ti0.3-Ag-8kV	77.2	21.1	1.3	0.4	0.0
PCL/PEO-Ti0.3-Ag-5kV	70.3	27.9	1.1	0.7	0.0
PCL-Cu	50.6	29.0	0.0	0.0	20.4
PCL-Cu-COOH	73.1	26.5	0.0	0.0	0.4

**Table 2 molecules-27-01333-t002:** The titers of the SARS-CoV-2 virus strain.

Sample	Virus Titer (LgCPE50)
PCL-Cu	1
PCL-Cu-COOH	1
PCL/PEO-Ti0.3-Ag-5kV	1,7

**Table 4 molecules-27-01333-t004:** Dimer binding energy calculated by DFT and ReaxFF, the energy difference between DFT and ReaxFF and bond length in angstroms.

Dimer	E, eV (DFT)	E, eV (ReaxFF)	ΔE, eV	R, Ǻ (DFT)	R, Ǻ (ReaxFF)
Ag-Ag	−2.17	−1.56	−0.61	2.56	2.64
Ag-C	−5.60	−4.63	−0.53	1.95	2.02
Ag-H	−6.77	−4.71	−0.41	1.62	1.48
Ag-O	−7.09	−5.63	−0.29	1.95	2.15

## Data Availability

Data is available upon a reasonable request.

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
