# Peer review of "Biodegradable Nanohybrid Materials as Candidates for Self-Sanitizing Filters Aimed at Protection from SARS-CoV-2 in Public Areas"

_molecules, 2022, doi:10.3390/molecules27041333_

Round 1

Reviewer 1 Report

I read an interesting and comprehensive research work entitled ‘Biodegradable nanohybrid materials as candidates for self-sanitizing filters aimed at protection from SARS-COV-2 in public areas’. The concept of the article is interesting and suitable to publish in Molecules. This manuscript is generally well written and clearly presented however still needs to address some comments, and thus require moderate revision.

  • Abstract looks very general and not informative should be rewritten. In abstract authors should mention the importance of research work briefly. Give results values.
  • In the introduction section, write the novelty of the work and the problem statement clearly. Add more details about the different approaches to tackle COVID pandemic and methodologies for exapmple refer Biosensors and Bioelectronics, 112969, 2021; Plants 10 (6), 1213, 2021. The detailed discussion about the novelty, significance of your research work and research gap relative to the literature is essential.
  • Have authors checked the stability of the produced materials?
  • For figure and table captions give all details which are quite expected. Don’t use any abbreviation.
  • Techno Economic challenges of the developed materials need to be addressed. What are the limitations and future research directions that need to be described by adding a new section before the conclusions section?
  • What are the limitation to use this methodology for commercial application ?.
  • The conclusion of the study needs to be added with the specific output obtained from the study, it could be modified with precise outcomes with a take home message. 
  • Some English and grammar mistakes are present that need to be correct to improve the quality of the manuscript.

Author Response

Review 1

We thank the reviewer for his/her very valuable comments and suggestions that helped to improve our paper. Below is poin-by-point answers to the questions that were addressed.

I read an interesting and comprehensive research work entitled ‘Biodegradable nanohybrid materials as candidates for self-sanitizing filters aimed at protection from SARS-COV-2 in public areas’. The concept of the article is interesting and suitable to publish in Molecules. This manuscript is generally well written and clearly presented however still needs to address some comments, and thus require moderate revision.

  • Abstract looks very general and not informative should be rewritten. In abstract authors should mention the importance of research work briefly. Give results values.

Answer: The answer was re-written accordingly and essential data were reported.

  • In the introduction section, write the novelty of the work and the problem statement clearly. Add more details about the different approaches to tackle COVID pandemic and methodologies for exapmple refer Biosensors and Bioelectronics, 112969, 2021; Plants 10 (6), 1213, 2021. The detailed discussion about the novelty, significance of your research work and research gap relative to the literature is essential.

Answer

The introduction was improved, and the recommended references were added.

  • Have authors checked the stability of the produced materials?

Answer.

In our previous work, Ref32, the stability of PCL-Cu and PCL-COOH in water and PBS was studied. The Cu release lasts ~ 24 hours.

  • For figure and table captions give all details which are quite expected. Don’t use any abbreviation.

Answer:

Figure captions were improved.

  • Techno Economic challenges of the developed materials need to be addressed. What are the limitations and future research directions that need to be described by adding a new section before the conclusions section?

Answer:

We have estimated the cost breakdown for the production of PCL-Cu surgical facemasks and added it in the discussion section. Our estimation was based on the prices for the all consumables delivered from China (as well as equipment) and the consumption rate and electrical consumption per piece for the Lab equipment. Based of our figures the cost for the PCL-Cu production (with a size of 18 cm by 10 cm) will be 0.28 USD per piece. Hence, the figures might be improved when the actual upscaling is made. Nevertheless, even this overestimated cost allows to build a positive business case with NPV and IRR from 790 000$ and 34%, respectively. All these estimations and discussions are added to the manuscript's revised version.

  • What are the limitation to use this methodology for commercial application ?

Answer:

There are no apparent significant limitations for the commercial application of such product, as high IRR and NPV provided in our estimations highlights.

For example there few exciting products, e.g.  the nanofibrous mask (although without antiviral coating) with a price 1$/piece

https://soomlab-korea.com/products/soomlab-hyper-purifying-breathing-mask-100pcs?variant=32154296844322

Nevertheless, the new production line setup will require the new equipment, location, and capital investments. In order to perform a successful business case, all the items must be acquired quickly, while the troubles in the value chains might significantly stretch the timeline of the project implementation. For example, the delivery of magnetrons and electrospinning will take from 1 year.

  • The conclusion of the study needs to be added with the specific output obtained from the study, it could be modified with precise outcomes with a take home message

Answer:

The conclusions were revised and new data related to the economic evaluation was added.

  • Some English and grammar mistakes are present that need to be correct to improve the quality of the manuscript.

Answer

The manuscript was thoroughly reviewed and all changes are highlighted.

Reviewer 2 Report

The author has reported the paper Biodegradable nanohybrid materials as candidates for sanitising filters aimed at protection from SARS-COV-2 in public areas. The following are the question to answer 1. How TiO2 and Ag+ implantation into PCL/PEO nanofibers biocompabitibility? 2. How this materials is applicable practically? 3. What is nature of surfactant? 4. Is there any effect of form and size of nanofibers on treatment of viral infection 5. Is carbon acts as stressing agent or killing agent? 6. Please updated the references. DOI: 10.46998/IJCMCR.2020.01.000009, DOI: 10.1039/C8NA00343B, DOI: 10.4236/abb.2019.1012032. 5.

Author Response

Review 2

We thank the reviewer for his/her very valuable comments and suggestions that helped to improve our paper. Below is poin-by-point answers to the questions that were addressed.

The author has reported the paper Biodegradable nanohybrid materials as candidates for sanitising filters aimed at protection from SARS-COV-2 in public areas. The following are the question to answer

  1. How TiO2 and Ag+ implantation into PCL/PEO nanofibers biocompabitibility?

Answer:

We did not report our cell viability results in this work, but of course, in our lab tests, we have studied the effect of Ag and TiO2 coatings on cell viability (fibroblasts). A proper combination of the parameters enhances the cell viability, whereas the coatings are (at least) not cytotoxic for all cases.

  1. How this materials is applicable practically?

Answer:

The nanofibers filters and facemasks  (without Ag and Cu coatings, although) are commercially available:

Price of the nanofibrous mask is 1$/piece

https://soomlab-korea.com/products/soomlab-hyper-purifying-breathing-mask-100pcs?variant=32154296844322

The addition of magnetron sputtering process will increase the price for the item, but the changes will not be dramatic and such masks may have the retail price around 1.5-2$/piece for upscaled process. 

We have estimated the cost breakdown for the production of PCL-Cu surgical facemasks and added it in the discussion section. Our estimation was based on the prices for the all consumables delivered from China (as well as equipment) and the consumption rate and electrical consumption per piece for the Lab equipment. Based of our figures the cost for the PCL-Cu production (with a size of 18 cm by 10 cm) will be 0.28 USD per piece. Hence, the figures might be improved when the actual upscaling is made. Nevertheless, even this overestimated cost allows building a positive business case with NPV and IRR from 790 000$ and 34%, respectively. All these estimations and discussion are added to the manuscript's revised version.

  1. What is nature of surfactant?

Answer:

In our process we did not use the surfactants. The deposition of Ag NPs and Cu was performed by ion implantation and magnetron sputtering, i.e. Physical Vapor Deposition process.

  1. Is there any effect of form and size of nanofibers on treatment of viral infection

Answer:

Thank you very much for  very interesting question. Indeed, this might be interesting to investigate in upcoming research. The morphology of nanofibers may affect the motion of the viruses and that enhance the killing process.

  1. Is carbon acts as stressing agent or killing agent?

Answer:

COOH coating acts as stressing agent, as well as the potential active center for possible drug immobilization to further enhance the antiviral activity.

  1. Please updated the references. DOI: 10.46998/IJCMCR.2020.01.000009, DOI: 10.1039/C8NA00343B, DOI: 10.4236/abb.2019.1012032. 5.

Answer:

Thank you so much for suggesting literature. We added the 10.1039/C8NA00343B (Ref25) while discussing the AgNPs.

Reviewer 3 Report

  1. There two main problems in this work. The first is the method used to detect the virus and its activity. The MTT tests is not a standard method to measure the antiviral property. The cell viability obtained by MTT assay can be influenced by virus infection or the cytotoxicity of the tested materials. The load of virus can be tested by TCID50 or PCR, which are both mature methods. The cell viability does not equate to viral load.
  2. The second is that there is no clear structure-to-property correlation. Although TiO2-Ag materials have been well characterized, they do not show any obvious antiviral property based on their method. Instead, for the copper materials that may have antiviral property against SARS-CoV-2, there is only a few structural information. How to control the copper ion release, and what is the critical concentration to have a noticeable antiviral property.

Minor parts:

  1. “Biodegradable” in the title has not been discussed in the manuscript.
  2. For 2.1, Simulation of Ag+ implantation into the PCL matrix, the simulation of Ag+ implantation into the PCL matrix was to provide adjustment suggestion for the design of material. However, the only one result derived from this experiment was that the average deviation angle for simulations was two times smaller in comparison with Cu atom, due to atomic weight difference. This result was obvious and provided no valuable suggestion. How to adjust the material design based on this result should be discussed further.
  3. For 2.2. Morphology of biodegradable nanohybrid material, the accelerating voltage for PCL/PEO-Ti0.5-Ag-15kv was 15 kV, while others were 2kv. Whether the destruction of nanofiber in PCL/PEO-Ti0.5-Ag-15kv was caused in the preparing step, or the SEM observation steps was not clear. This part should be reconfirmed.
  4. Many content in the main text should be briefly be summarized, e.g. line 59-74, 75-93, 101-115, 125-150, 360-383. This is not a review paper!
  5. Please show the power equation for Cu penetration to PCL in the article instead of putting in the reference solely. The two abbreviations PCL (Polycaprolactone) and PEO (Polyethylene oxide) are used throughout the article, and their full names need to be added. The formula of in sentence 304 is garbled. The scale bar in Figure 4 and Figure 11 should be added.

Author Response

We thank the reviewer for his/her very valuable comments and suggestions that helped to improve our paper. Below is point-by-point answers to the questions that were addressed. The PDF file with full answers with images is attached.

  1. There two main problems in this work. The first is the method used to detect the virus and its activity. The MTT tests is not a standard method to measure the antiviral property. The cell viability obtained by MTT assay can be influenced by virus infection or the cytotoxicity of the tested materials. The load of virus can be tested by TCID50 or PCR, which are both mature methods. The cell viability does not equate to viral load.

Answer

We agree that TCID50 (defines 50% cell inghibition dose of viruses) assay standard method to measure the antiviral property. In our work, we used TCID50 assay.

We do apologize that the methodology was not adequately and fully described. The text is nowcorrected and Table 2 with virus titers is provided.

The TCID50 is the dose of a certain virus required to kill 50% of the cells in a well compared to uninfected control. This technique allows for more rapid quantification of the virus infectious units present in a sample compared to the plaque/immunofocus assay.  In a traditional TCID50 assay, a serial dilution of the sample to test is incubated with cells for enough time to generate quantifiable cell death. The TCID50 assays are run in 96-well plates, and the readout is MTT uptake for cell death.

.

  1. The second is that there is no clear structure-to-property correlation. Although TiO2-Ag materials have been well characterized, they do not show any obvious antiviral property based on their method. Instead, there is only a few structural information for the copper materials that may have antiviral property against SARS-CoV-2. How to control the copper ion release, and what is the critical concentration to have a noticeable antiviral property.

We thank the reviewer for this valuable comment, but would like also to mention that XPS, and WCA were used to characterized PCL-Cu and PCL-COOH surfaces, while SEM micrographs and Cu ions released was referenced to our previous work (Ref32) that is published in the open-access Membranes journal. Hence, all information related the the structure of PCL-Cu and PCL-Cu-COOH samples is provided. We also added the techno-economic assessment of proposed technology and provided the cost analysis, NPV and IRR.

We would like also to highlight our findings can be very useful from practical and scientific points of view, because in the majority of analyses, the Ag and TiO2-based nanomaterials are used. Here, the negative results for the SARS-COV2 for the PCL/PEO-Ti-Ag samples highlight the complexity of the antiviral activity problem of nanomaterials, which is less studied in contrast to antibacterial nanomaterials.

Minor parts:

  1. Biodegradable” in the title has not been discussed in the manuscript.

Answer:

As our materials are based on the biodegradable PCL and PCL/PEO nanofibers, we considered that the resulting bionanohybrid material would also be biodegradable in the compost conditions. The amount of Cu, Ag and Ti is very minor and these ions will play an important role in the metabolism, we assumed that out material will be biodegradable. Hence, the utilization of the disposed of facemasks will not require incineration.

  1. For 2.1, Simulation of Ag+ implantation into the PCL matrix, the simulation of Ag+ implantation into the PCL matrix was to provide adjustment suggestion for the design of material. However, the only one result derived from this experiment was that the average deviation angle for simulations was two times smaller in comparison with Cu atom, due to atomic weight difference. This result was obvious and provided no valuable suggestion. How to adjust the material design based on this result should be discussed further.

Answer:

The ion implantation at a low energy range (from 500 eV to 2400 eV) was studied at the atomic level via classical molecular dynamics simulations. This will allow one to tune ion implantation process more carefully.

  1. For 2.2. Morphology of biodegradable nanohybrid material, the accelerating voltage for PCL/PEO-Ti0.5-Ag-15kv was 15 kV, while others were 2kv. Whether the destruction of nanofiber in PCL/PEO-Ti0.5-Ag-15kv was caused in the preparing step, or the SEM observation steps was not clear. This part should be reconfirmed.

Answer:

Unfortunately, we did not have the SEM for PCL/PEO-Ti-Ag15kV at 2kV. Nevertheless, according to our experience with nanofibers, we never observed any changes in the structure of the nanofibers under SEM gun. You may find our SEM images of different nanofibers at 2, 10, 15 and even 20 kV. Indeed, the changes are related to the intrinsic structure of the sample. To confirm this, please see the SEM micrograph of PCL-Ti-Ag15kV sample at 2kV (below).

Unfortunately a higher magnification picture was taken at 15 kV.

  1. Many content in the main text should be briefly be summarized, e.g. line 59-74, 75-93, 101-115, 125-150, 360-383. This is not a review paper!

Answer:

We revised the text substantially, but we would like to keep some parts here, as this interdisciplinary work is quite complex and readers from different research topics can be interested in our work. Finally, our manuscript is not very lengthy for a regular research paper (20 pages).

  1. Please show the power equation for Cu penetration to PCL in the article instead of putting in the reference solely. The two abbreviations PCL (Polycaprolactone) and PEO (Polyethylene oxide) are used throughout the article, and their full names need to be added. The formula of in sentence 304 is garbled. The scale bar in Figure 4 and Figure 11 should be added.

Answer:

We have added the power equation for Cu penetration to PCL

Round 2

Reviewer 1 Report

I carefully read the revised version of the manuscript. Authors answered well for all raised comments thus the present form of the manuscript can be accepted.

Congratulations !!